# All-optical switching based on plasmon-induced Enhancement of Index of Refraction

Rakesh Dhama [1], Ali Panahpour [1], Tuomas Pihlava [1], Dipa Ghindani [1] & Humeyra Caglayan [1✉]

In quantum optical Enhancement of Index of Refraction (EIR), coherence and quantum interference render the atomic systems to exhibit orders of magnitude higher susceptibilities with vanishing or even negative absorption at their resonances. Here we show the plasmonic analogue of the quantum optical EIR effect in an optical system and further implement this in a linear all-optical switching mechanism. We realize plasmon-induced EIR using a particular plasmonic metasurface consisting of a square array of L-shaped meta-molecules. In contrast to the conventional methods, this approach provides a scheme to modulate the amplitude of incident signals by coherent control of absorption without implementing gain materials or nonlinear processes. Therefore, light is controlled by applying ultra-low intensity at the extreme levels of spatiotemporal localization. In the pursuit of potential applications of linear all-optical switching devices, this scheme may introduce an effective tool for improving the modulation strength of optical modulators and switches through the amplification of input signals at ultra-low power.

[1] Tampere University, Faculty of Engineering and Natural Sciences, 33720 Tampere, Finland. ✉email: humeyra.caglayan@tuni.fi

Ultra-compact optical switches and modulators with ultra-short response time and minimal energy consumption are highly desirable in nanophotonics for the development of efficient all-optical computing, photonic information processing, and networking[1]. Conventional all-optical nanoswitches are mainly based on nonlinear effects, such as Kerr[2], Raman[3], two-photon absorption[4] and frequency mixing[5]. To improve the performance of nonlinear optical switches, they are usually equipped with plasmonic or resonant nanostructures to exploit their field enhancement and confinement features[6–9]. Other switching schemes involve the hybridization of metamaterials by functional components such as semiconductors, carbon nanotubes, graphene, liquid crystals, and phase-change materials[10–15] and a metamaterial of plasmonic circuits[16]. However, the aforementioned switching mechanisms generally suffer from unsatisfactory response time and/or high power requirements.

Linear interference phenomena are introduced as a promising approach to achieving ultrafast all-optical switching[16–18] requiring arbitrarily low-intensity optical beams down to the level of single-photon regime[19]. The realization of this approach is based on the interference of two collinear, counter-propagating and phase-controlled coherent beams interacting with a plasmonic metasurface of subwavelength dimensions. The absorption of standing wave light is modulated by controlling the phase or amplitude of one of the counter-propagating light waves and consequently tuning the position of the nodes and antinodes on the metasurface[20,21].

As another more efficient approach, a class of linear optical nanoswitches and modulators are specifically designed to meet the requirements of the coherent perfect absorption (CPA) phenomenon[22,23], which is a time-reversed counterpart to laser emission and a generalization of the concept of critical coupling to an optical cavity[24]. In a CPA system, complete absorption of electromagnetic radiation is achieved by controlling the interference of multiple incident waves. It can be realized in a variety of photonic structures, including planar and guided-mode structures, graphene-based systems, parity- and time-symmetric structures, epsilon-near-zero multi-layer films, quantum-mechanical systems and chiral metamaterials[25–27]. In a plasmonic nanostructure, CPA occurs when coherent light is completely absorbed and transferred to surface plasmons by exciting the nanostructure with the time-reversed mode of the corresponding surface plasmon laser or the so-called SPASER[23]. In experimental realizations of the effect, usually two symmetric plane waves incident on opposite sides of the system are completely absorbed, as a result of critical coupling into the dissipative degrees of freedom of the system.

The plasmonic counterparts of some quantum optical effects have also been widely implemented to develop the functionality of optical systems, metamaterials and switching devices[24,28,29] including the Fano resonance effect and electromagnetically induced transparency (EIT). In plasmonic systems, the plasmon-induced transparency (PIT) creates a narrow transparent window within a broader absorption band of the system due to destructive interference of super-radiant (radiative) and subradiant (dark) resonance modes of the plasmonic nanostructures. This effect is not only limited by the narrow band but also by its weak response. Typically, in an optical switch based on PIT, the transparency window of the medium is reversibly shifted in a wavelength range to provide the ON and OFF states of the switch by applying a light-induced change in the refractive index of a nonlinear material integrated with plasmonic constituents[15,17]. Furthermore, a large resonant index of refraction with vanishing absorption has also been reported in quantum optics. This fascinating phenomenon arises when coherence takes place between an excited state and coherently prepared ground state doublet of atomic gas and is known as an enhancement of index of refraction (EIR)[30]. Recently, plasmonic analogue of this quantum optical EIR is theoretically reported in coupled plasmonic resonators, enabling the maximum susceptibility (strong electromagnetic response) with zero optical losses at resonance frequencies via coherent control of surface plasmons[31]. Thus, apart from some newly proposed potential applications[32,33], this approach has suggested a unique path to realize loss-compensated plasmonic devices operating at resonance frequencies through extraordinary enhancement of refractive index without using any gain media[34–37] or nonlinear processes[28,38].

In this work, we introduce and experimentally realize an ultrafast all-optical switching mechanism based on the plasmonic analogue of the EIR effect by designing and fabricating a particular plasmonic metasurface consisting of a square array of L-shaped meta-molecules. In contrast to the conventional methods of optical switching between zero and complete absorption limits of the system, our scheme is based on a linear phenomenon that enables the switching of system absorption between positive and negative values at low exciting power levels. This is achieved through coherent control of polarizability of nanoantennas by controlling the phase and amplitude of a control beam with a polarization perpendicular to the signal beam resulting in unprecedented control of modulation strength of optical switches. Therefore, the scheme is not restricted to counter-propagating light waves and the system can be illuminated by the signal and control beams from one side. Our approach also shows how the absorption in the plasmon resonance band of metasurface can be controlled as per demand using the properties of the control beam as a tool.

## Results and Discussion

**Analytical model.** In this section, an analytical model is presented to describe the proposed switching mechanism and estimate the performance of the plasmonic metasurface acting as a nanoswitch. Here, a meta-molecule consisting of two perpendicular nanoantennas in $x$ and $y$ directions is illuminated by signal and control beams propagating along the $z$ direction. For simplicity of analytical calculations, we model the constituent nanoantennas of the meta-molecule as identical spheroidal nanoparticles (NPs). Since the dimensions of nanoantennas are much smaller than the wavelength in our working spectral range, dipole approximation is sufficient and there is no need for considering higher-order multipolar resonances beyond dipolar. Furthermore, electrostatic approximation can be used for evaluating the polarizability of nanoantennas and the description of the EIR effect as demonstrated in ref. [31]. This approximation accounts for static depolarization through geometrical parameter L and also phase retardation due to the Ohmic loss of metallic nanoantennas, represented by the gamma constant in the Drude formula. However, here we apply the improved quasistatic approximation for achieving better compliance with simulation and experiment. In addition to the abovementioned depolarization and phase retardation effects, this approximation also accounts for radiative phase retardation due to light scattering by the nanoantennas, represented by $k^3$ depolarization term in the following relations, which affects the linewidth of surface plasmon resonances and consequently the overall resonance profile of the coupled NPs. We ignored the nonradiative $k^2$ depolarization term because it just produces a resonance frequency redshift that does not improve the conformity with simulation and experimental results.

The polarizability of each NP in the quasi-static approximation is given by:

$$\alpha^{-1} = \alpha_0^{-1} - i\frac{2k^3}{3} \qquad (1)$$

in which $-2ik^3/3$ is the radiation correction term for the NP's electrostatic polarizability $\alpha_0$ and $k$ is wavenumber. The electrostatic polarizability for each NP in free space is given by:

$$\alpha_0^{-1} = \frac{1}{\nu}\left(L + \frac{1}{\epsilon - 1}\right), \tag{2}$$

where $\nu$ and $\epsilon(\omega)$ are the volume and frequency-dependent dielectric function of NPs respectively, and $L = \frac{1-e^2}{e^2}\left(-1 + \frac{1}{2e}\ln\frac{1+e}{1-e}\right)$ is an electrostatic geometrical factor in terms of eccentricity $e = \sqrt{1 - b^2/a^2}$ of the spheroids with $a$ and $b$ as their respective longer and shorter semi-axes. Using the Drude formula $\epsilon = 1 - \omega_p^2/\omega(\omega + i\gamma)$ for dielectric function of the NPs and normalizing the angular frequency and relaxation constant to plasma frequency as $\tilde{\omega} = \omega/\omega_p$ and $\tilde{\gamma} = \gamma/\omega_p$, from relations (1) and (2) we obtain:

$$\alpha^{-1} = \frac{1}{\nu}(L - \tilde{\omega}^2 - i\tilde{\gamma}\tilde{\omega}) - i\frac{2k^3}{3} \tag{3}$$

The structure is illuminated by an $x$-polarized "signal" beam and a $y$-polarized "control" beam of the same frequency but fourfold larger amplitude. Assuming geometrical parameters of $a = 70$ nm and $b = 8.4$ nm ($L = 0.0267$) for the NPs and center to center distance of $R = 90$ nm, the real and imaginary parts of polarizability of the nanoantenna along $x$ direction, probed by the $x$-polarized signal are shown in Fig. 1(a) as a function of normalized frequency. The curves correspond to the conditions when the control beam has the phase difference ($\delta\phi$) values of $-\pi/2$ and $\pi/2$ with respect to the signal beam. The curves show the extinction of the probe beam (which is proportional to the imaginary part of polarizability) taking positive or negative values depending on the phase of the control beam. It is also interesting that the extinction is zero at wavelengths near extreme points of real polarizability. This particular profile of polarizability corresponds to the EIR phenomenon in contrast to the EIT effect, where the medium shows minimal extinction at the zero polarizability point.

To study the collective response of the meta-molecules as an optical nanoswitch, we consider a metasurface consisting of a square array of such meta-molecules. In this two-dimensional array, the induced dipole moment of each nanoantenna in $x$ or $y$ directions ($p_{x,y}$) is a result of contributions from external field components along the respective $x$ and $y$ directions ($E_{x,y}$), as well

as the electric field due to the closely coupled orthogonal nanoantenna (with coupling coefficient $C$), and finally the fields due to the collective effect of other parallel ($S_\parallel p_{x,y}$) and perpendicular ($S_\perp p_{x,y}$) nanoantennas in terms of the corresponding lattice sums $S_\parallel$ and $S_\perp$:

$$p_x = \alpha(E_x + Cp_y + S_\parallel p_x + S_\perp p_y) \tag{4}$$

$$p_y = \alpha(E_y + Cp_x + S_\parallel p_y + S_\perp p_x) \tag{5}$$

which can be represented in matrix form as:

$$\begin{pmatrix} p_x \\ p_y \end{pmatrix} = \begin{pmatrix} \alpha_{xx} & \alpha_{xy} \\ \alpha_{xy} & \alpha_{yy} \end{pmatrix}\begin{pmatrix} E_x \\ E_y \end{pmatrix}, \tag{6}$$

where:

$$\hat{\alpha} = \begin{pmatrix} \alpha_{xx} & \alpha_{xy} \\ \alpha_{xy} & \alpha_{yy} \end{pmatrix} = \frac{\alpha'}{1 - C'^2\alpha'^2}\begin{pmatrix} 1 & C'\alpha' \\ C'\alpha' & 1 \end{pmatrix}. \tag{7}$$

Here $\alpha' = \alpha/(1 - \alpha S_\parallel)$, $C' = C + S_\perp$ and the coupling coefficient $C = 3/8\pi\epsilon_0 R^3$ represent the near field interaction of coupled nanoantennas with center to center distance $R$. We note that while the polarizability of each NP is affected by the radiated fields of all other parallel NPs, the field due to the perpendicular NPs just modifies the coupling constant $C$. It can be shown that the dipolar fields due to the perpendicular NPs symmetrically located on either side of each nanoantenna neutralize the effect of each other. Therefore, the contribution of $S_\perp$ can be neglected for a metasurface consisting of large number of meta-molecules. For normal incidence on the metasurface with the dense array of meta-molecules ($kd < 1$), the interaction constant $S_\parallel$ representing the effect of all dipoles parallel to a specific nanoantenna is given by[39]:

$$S_\parallel \approx -i\frac{\omega}{A}\frac{\eta}{4}\left(1 - \frac{1}{ikD}\right)e^{-ikD} \tag{8}$$

Here, $A = d^2$ is the unit cell area, $\eta = \sqrt{\mu_0/\epsilon_0}$, $k = \omega\sqrt{\mu_0\epsilon_0}$ and $D = d/1.438$.

Finally a transmission matrix of the form:

$$\begin{pmatrix} E_{tx} \\ E_{ty} \end{pmatrix} = \begin{pmatrix} T_{xx} & T_{xy} \\ T_{xy} & T_{yy} \end{pmatrix}\begin{pmatrix} E_{ix} \\ E_{iy} \end{pmatrix} \tag{9}$$

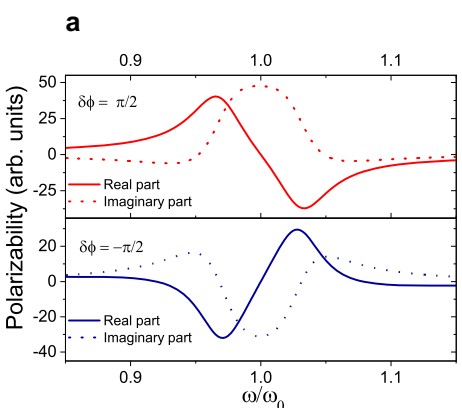

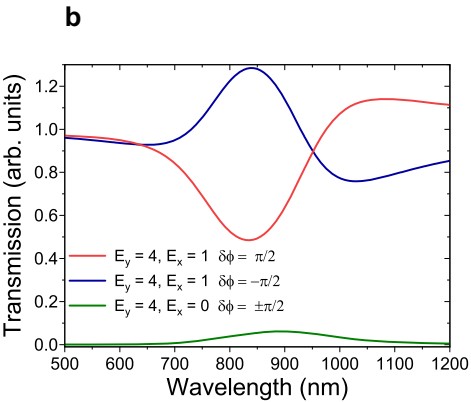

**Fig. 1 Polarizability of coupled spheroidal nanoparticles and calculated transmittance of the meta-molecules. a** Real (solid lines) and imaginary (dotted lines) parts of polarizability of a spheroidal NP probed by a signal beam while nearly coupled to another orthogonally oriented identical NP which is excited by a phase correlated control beam. The control beam has either $\pi/2$ (upper curves) or $-\pi/2$ (lower curves) phase difference with respect to the signal beam. Geometrical parameters of $a = 70$ nm and $L = 0.0267$ are used for the NPs with center to center distance of $R = 90$ nm. **b** The collective transmission response of the meta-molecules corresponding to $\delta\phi = \pm\pi/2$ phase differences (blue and red curves) of the control beam. The green curve shows the leakage of control beam to the signal channel when signal is zero.

can be defined with the matrix components[40]:

$$T_{xx} = T_{yy} = 1 + \frac{ik\alpha_{xx}}{2\epsilon_0 d^2}, \quad T_{xy} = -\frac{ik\alpha_{xy}}{2\epsilon_0 d^2} \qquad (10)$$

to calculate the transmission through the metasurface.

Assuming the same geometrical parameters as in Fig. 1(a) and unit cell dimension of $d = 320$ nm, we obtain the signal transmission curves shown in Fig. 1(b) for signal and control beam amplitudes of $E_{signal} = 1$ and $E_{control} = 4e^{\pm i\delta\phi}$, respectively. For the sake of simplicity, the effect of substrate is ignored in the analytical calculations. We note that the maximum transmission can be higher than 100% representing signal amplification. The amplification strength depends on the phase and amplitude of the control beam. The extinction or amplification of input signal in different spectral regions of the transmission curves are correlated with and directly proportional to the imaginary part of the polarizability of nanoantennas along the $x$ direction. In spectral ranges with a negative imaginary part of polarizability, amplification of signal occurs and in contrast, in the regions with positive values of imaginary polarizability, signal extinction is observed. The strength of amplification or extinction is proportional to the magnitudes of the respective negative or positive values of the imaginary part of the polarizability. This amplification feature provides the capability of improving the modulation strength of the switch. In Fig. 1(b), the transmission of the control beam through the signal channel is also plotted (the green curve) when there is no input signal. This leakage should be reduced to a small fraction of the modulation strength.

**Experimental Results and Simulations**. The plasmonic meta-surface has been realized by fabricating two identical gold nanorods of dimensions $130 \times 40 \times 30$ nm³ (length × width × height) as L-shaped meta-molecules with a period of 320 nm on a fused silica substrate by electron beam lithography process. Scanning electron microscopy (SEM) images of the fabricated metasurface are shown in Fig. 2(a) visualizing the pattern of two closely placed orthogonal nanorods. The inset of Fig. 2(a) indicates the dimensions of the nanorods which are nearly identical to the values used in our Finite-difference time-domain (FDTD) simulations.

Figure 2 (b) illustrates the intrinsic response of the metasurface in the absence of phase delay and polarization control. Experimental (solid orange line) and simulated (solid olive green line) transmission spectra of the metasurface show the induced plasmon resonances around 800 nm under the illumination of broadband, incoherent and unpolarized light source. A slight redshift in the experimental transmission spectrum with respect to the simulated one can be attributed to minor fabrication imperfections, particularly the edges of nanorods.

Before investigating the switching performance, we have confirmed the contribution of unwanted leakage of the control beam into the signal channel through simulations as well as experiments. To identify the contribution of $y$-polarized control beam in the $x$-polarized output signal channel, the meta-structure is illuminated with a control beam ($E_y = 1$) excitation without any signal beam ($E_x = 0$). Leakage is the measure of purity of the signal and a minimal value of this contribution is an important factor for the efficient performance of optical switches and modulators. We have obtained around 4% leakage of the control beam to the signal channel as demonstrated in Fig. 2(c).

To demonstrate the potential of this metasurface as an all-optical nanoswitch, Fig. 3(a) represents the schematic for the used optical configuration. A broadband incoherent light beam travels through a linear polarizer and a phase delay component (quarter-wave plate) to generate phase and polarization controlled signal and control beams and excites the metasurface. The incident beam on each meta-molecule of the metasurface can be described as two input beams that are distinguished by their orthogonal polarizations. A horizontally polarized beam along the $x$ direction is entitled a 'signal beam', while a vertically polarized beam along the $y$ direction is called a 'control beam' as presented in Fig. 3(a). Another linear polarizer placed after the metasurface in the transmitted path ensures the removal of any unwanted control beam component in the output and only allows the output signal beam to the detector. The ratio of the amplitudes of control beam to signal beam is defined as $C_{amp}$, which is set by changing the input polarizer angle. For each $C_{amp}$ value, two measurements of $\delta\phi = \pi/2$ and $\delta\phi = -\pi/2$ are performed by maintaining the $C_{amp}$ constant.

The experimental realization of this switching effect is demonstrated in Fig. 3(b) with an excitation beam of $C_{amp} = 4$ and $\delta\phi = \pm\pi/2$. After transmitting through the metasurface, the beam passes through a linear polarizer parallel to the $x$-axis as shown in Fig. 3(a). The experimental procedure to ensure the desired electric field ratio between control and signal beams ($C_{amp}$) and the alignment of linear polarizers used in excitation and transmission paths have been discussed in detail in the Methods section. Figure 3(b) clearly demonstrates the strong switching effect (blue curve over red one) by inducing the giant transmission enhancement in the whole plasmon resonance band supported by the L-shaped meta-molecules when the control beam phase lags behind the signal beam with a phase difference $\delta\phi = -\pi/2$ and enables modulation of output signal intensity to over 200% $x$-polarized transmittance.

We note that coherent laser sources are not necessarily required to achieve coherent control of surface plasmons. Modulation strictly depends on $C_{amp}$ and phase difference between control and signal beams which must be coherent with respect to each other, in spite of originating from an incoherent light source. It is also noteworthy that such an all-optical switching phenomenon can not be achieved when two separate incoherent sources are employed for signal and control beams due to the absence of mutual coherence.

Next, the $x$-polarized output signal intensity was numerically calculated when the metasurface was excited with a $y$-polarized control beam with amplitude $C_{amp} = E_y/E_x = 4$ along with a phase difference $\delta\phi = \pm \pi/2$. Remarkable enhancement in signal output intensity was numerically acquired with a phase difference of $\delta\phi = -\pi/2$ with respect to $\delta\phi = +\pi/2$ at the plasmon resonance band of the metasurface as seen in Fig. 3(c). This predicts strong modulation in the signal beam intensity regulated by the phase change of the control beam. A detailed description of the performed FDTD simulations is included in the Methods section.

The extraordinary enhancement in output signal intensity ($x$-polarized light) is attributed to electromagnetically induced negative absorption in the horizontal nanoantenna via coherent control of surface plasmon resonances and this mechanism is different compared to previous work[16] which is based on the interference and depends on the angle of incidence, and the modulation is only observed for non-normal incidence. In order to understand the physical mechanism involved in this outstanding effect, we must revisit a classical analogue of the EIR effect in quantum optics, which states that atoms prepared in coherent superposition state can induce a large resonant index of refraction with vanishing absorption via quantum interference between an excited state and coherently prepared ground state doublet[30]. In the plasmonic analogue of the EIR effect, two perpendicular metallic nanorods are excited by two orthogonally polarized fields with the phase difference $\delta\phi = \pi/2$ resulting in low-loss localized surface plasmon resonance (LSPR) of a nanoantenna. When both nanoantennas of the L-shaped meta-molecule are coupled under the illumination of the incident light beam and fulfill the mutual coherence condition,

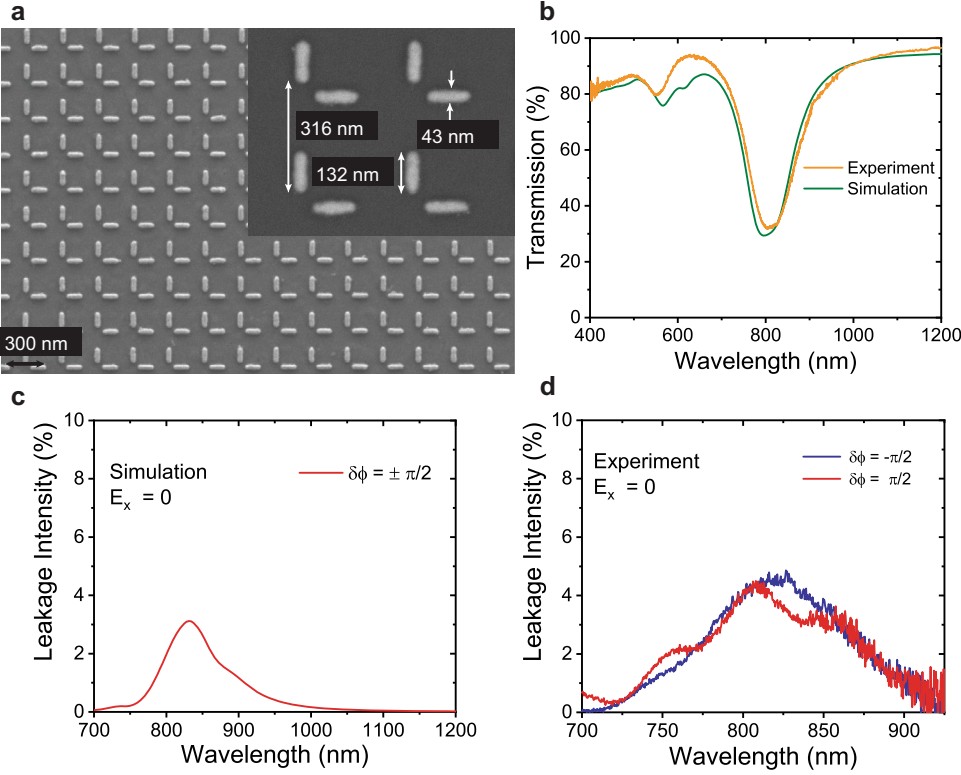

**Fig. 2 Metasurface's design realization, spectral response and leakage of the control beam. a** SEM image of the fabricated L-shaped metasurface, the inset indicates the nanorod dimensions and period of the metasurface. **b** Experimental and calculated transmission spectra of the metasurface without any phase delay or polarization control. **c**, **d** The fraction of control beam leaking in the output signal intensity, when signal is zero.

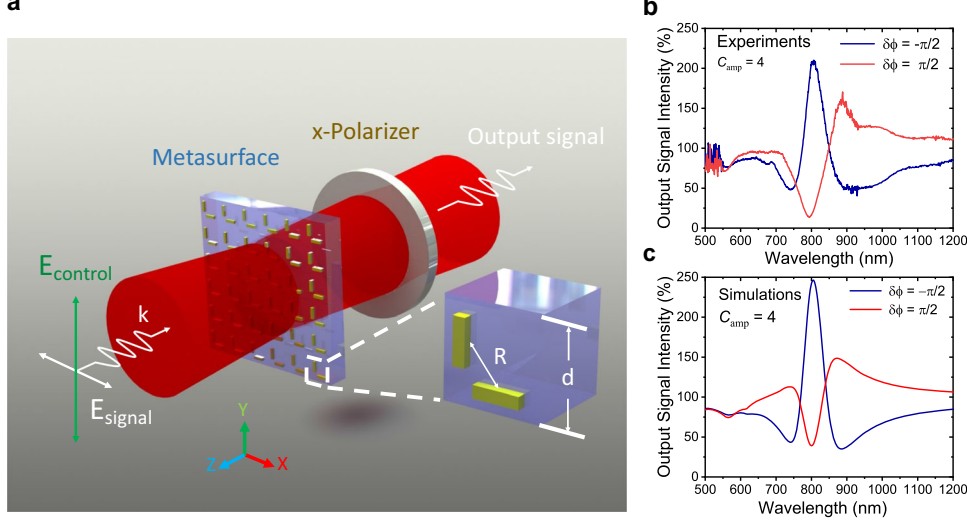

**Fig. 3 All-optical nanoswitch based on coherent control of surface plasmons. a** Schematic illustration for the modulated output signal intensity of the device under the excitation of an *x*-polarized signal beam and a phase-delayed *y*-polarized control beam from a broadband light source. The inset of the figure represents the schematic of a single L-shaped meta-molecule as a unit cell of the metasurface with a period of ($d$) and center to center distance ($R$) of nanorods. **b**, **c** Experimental and simulated modulated output signal intensity at $C_{amp}$ = 4 with phase differences of ± $\pi$/2 between control and signal beam, demonstrating the strong switching effect.

then surface plasmons excited in the vertical nanoantenna (control) induce an extraordinary modification in the resonance profile of the horizontal nanoantenna probed by the *x*-polarized signal beam. This modification of the spectral profile of LSPR and induced zero or negative absorption is attributed to the mutual coupling of both nanoantennas and the canalization of energy from vertically (control beam) to horizontally (signal beam) oriented nanoantenna in the

presence of specific phase difference between signal and control beams. Thus, this energy exchange enhances the output signal intensity and enables the all-optical switching effect. In order to verify the canalization of energy from control to signal beam, a decrease in transmission of control beam has been numerically as well as experimentally shown with the phase difference $\delta\phi = \pi/2$ in Supplementary Fig. 1.

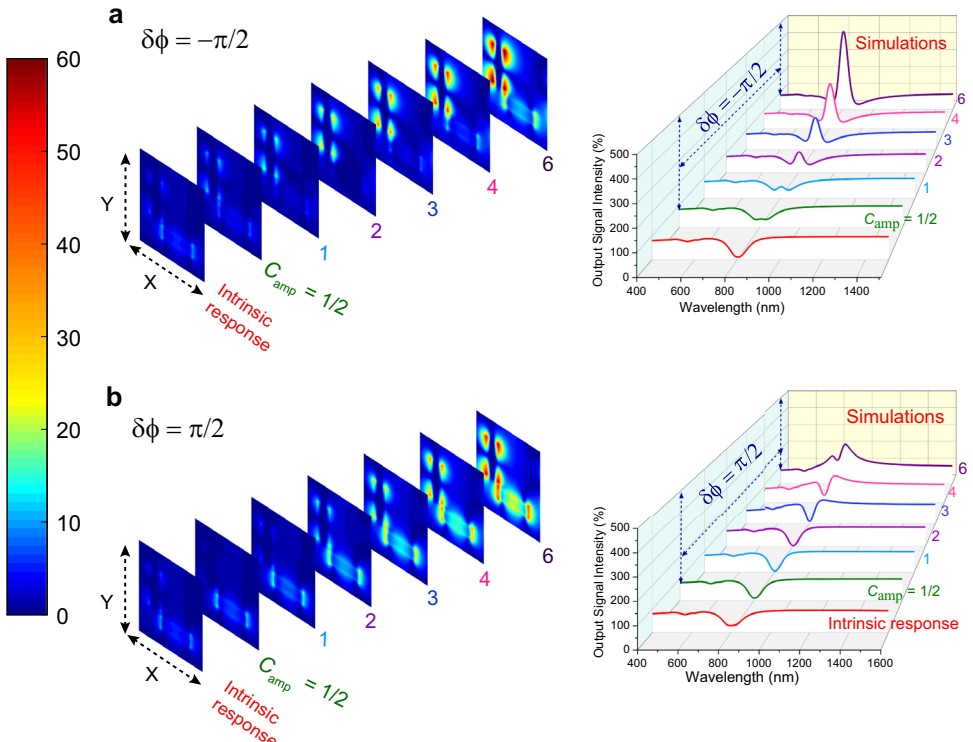

**Fig. 4 Electric field profiles and spectral response of output signal intensity. a, b** *x*-polarized electric field profiles and calculated output signal intensity curves as a function of control beam amplitude $C_{amp}$ with phase difference of $\delta\phi = -\pi/2$ and $\delta\phi = \pi/2$ in comparison to the intrinsic response, respectively.

Furthermore, in order to clearly understand the index enhancement phenomenon in an L-shaped meta-molecule, we have performed E-field simulations and numerically calculated the output signal intensity with the changes in the phase difference and amplitudes of the control beam. In this regard, Fig. 4(a) shows 3D waterfall plotting of simulated output signal intensity curves and their corresponding field profiles as the function of $C_{amp}$ with phase difference of $\delta\phi = -\pi/2$, while Fig. 4(b) reports the similar curves and field profiles for the phase difference of $\delta\phi = \pi/2$. Note, that the intrinsic response of the structure refers to the absence of any phase delay component and polarizer in the optical configuration. When the control beam phase lags behind the signal beam with a phase difference $\delta\phi = -\pi/2$, the coherent interaction enables the modulation of the output signal. While there is no modulation in output signal intensity at resonance wavelengths with the phase difference $\delta\phi = \pi/2$ due to the absence of coherence in the system. More specifically, *x*-polarized field profiles with phase difference of $\delta\phi = -\pi/2$ exhibit very weak field confinement (see Fig. 4(a)) with respect to their corresponding field profiles with phase difference of $\delta\phi = \pi/2$ as shown in Fig. 4(b). Although the control (*y*-polarized) and signal (*x*-polarized) beam intensity has been defined through the same $C_{amp}$ parameter, the results show that horizontal (*x*-oriented) metallic nanorod behaves as a high index dielectric and is unable to confine strong field like metallic nanostructures under coherent control of plasmons, which is definitely clear evidence of index enhancement.

Finally, to demonstrate the transmission enhancement of signal beam in the plasmon band of the metasurface experimentally, we obtained output signal intensity curves as the function of $C_{amp}$ with phase difference of $\delta\phi = -\pi/2$ in comparison to the intrinsic response of the metasurface (Fig. 5(a)). The simulated output signal intensity curves clearly show a trend in the transmission enhancement as shown in Fig. 5(b) with the increase in the values

of $C_{amp}$ as compared to intrinsic plasmon band (red solid line) of this metasurface. Furthermore, Fig. 5(b) demonstrates that the transmission enhancement in the plasmon band of the metasurface (upward dotted black arrow) emerges at the cost of enhanced absorption in both arms (downward dotted black arrow) of output signal intensity curves. This behaviour can be explained by the causal nature of the response of metasurface via Kramers–Kronig dispersion relations. This is also in analogy with a gain-plasmonic system where the reduction of absorption in the selective region of the plasmon band leads to an increase in the absorption in its neighbouring wavelength band due to the decrease in the imaginary part of permittivity via exciton-plasmon coupling effect[41]. In a similar way, when the imaginary part of permittivity is reduced due to the EIR effect, transmission enhancement in the plasmon band of metasurface leads to a decrease in the transmission on both sides of transmission curves of output signal intensity.

This index enhancement effect via coherent control of surface plasmons can be used as a technique to control absorption (optical losses) in the plasmon resonance band without using any gain media. In the plasmon analogue of the EIR effect, since both of the coupled resonances are bright states with rather large linewidths, the induced transparency has also a large bandwidth which can be further increased by enhancing the coupling strength of the two nanoantennas. Our experimental and simulated results clearly show that this technique creates broad transparency almost in the whole plasmon band of the metasurface with full width at half maximum (FWHM) of 100 nm as shown in Supplementary Figure 2 of SI. Indeed, this approach can be implemented across the entire visible and near-infrared region to minimize the optical losses using different metal nanostructures and specifically engineered designs. On the other hand, controlling the absorption through such metasurfaces can realize novel devices such as variable attenuators, coherence filters and

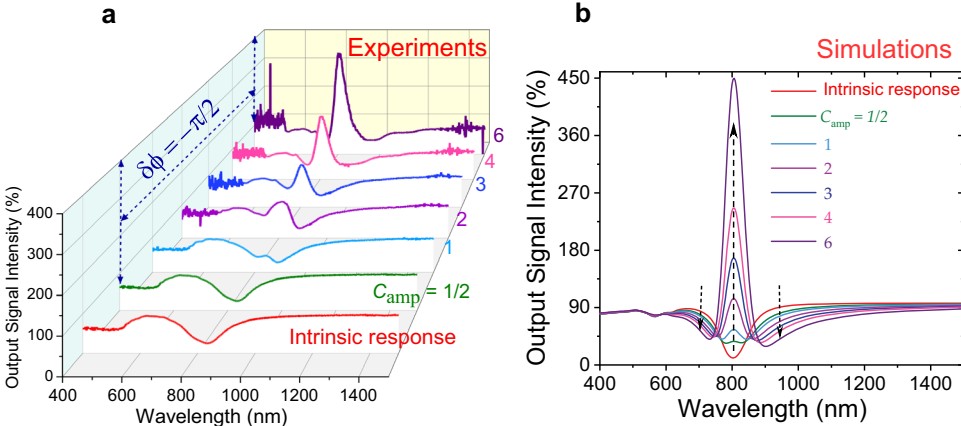

**Fig. 5 Broadband transmission enhancement of the signal beam in the plasmon resonance band of the metasurface. a** Experimental output signal intensity curves as a function of $C_{amp}$ with phase difference of $\delta\phi = -\pi/2$ in comparison to the intrinsic response of the metasurface. **b** 2D representation of the transmitted output signal intensity curves as a function of the amplitude of the control beam ($C_{amp}$) with a phase difference of $\delta\phi = -\pi/2$ in comparison to the intrinsic response of the metasurface when there is no polarizer and phase delay component included in the optical system.

coherent modulators, operating at arbitrarily low power levels[17]. Furthermore, it should be noted that the switching mechanism is not restricted to the phase differences of $\pm\pi/2$ between signal and control beams. Other phase differences may also result in significant modulation strengths, useful for switching applications, as can be seen in Supplementary Figs. 3–5. However, the maximum modulation strength may occur at slightly different wavelengths when phase differences are changed to other values.

In conclusion, we have reported an experimental demonstration of the EIR effect in a plasmonic system. An all-optical switching mechanism based on the plasmonic counterpart of the quantum optical EIR effect has been introduced. We realized this by utilizing phase correlated orthogonally polarized optical control and signal beams interacting with a metasurface consisting of L-shaped metamolecules. The switching mechanism is described using a simple analytical model as well as FDTD simulations and the fabricated metadevice characterized under different control beam intensities to identify the modulation. In this switching method, the absorption and dispersion of LSPR of the constituent nanoantennas are coherently and linearly controlled. Thus, extinction of the plasmonic system probed by the input signal in one polarization direction can take positive or negative values depending on the phase of the control beam. This enabled all-optical switching via plasmonic metasurface with 200% modulation strength. This switching scheme provides significant improvement in modulation strength with low-intensity control beams. The conventional methods of optical modulation and switching are based on coherent perfect absorption, requiring counter-propagating light waves. In contrast, this presented approach provides a simple experimental method of controlling light by light at the extreme levels of spatiotemporal localization by applying ultra-low intensity pulses at the level of single-photon to single nanoscale plasmonic meta-molecules.

## Methods
**Fabrication Procedure.** The fabrication process of the L-shaped metasurface was initiated by spin-coating a 120 nm thick layer of positive resist poly(methyl methacrylate) (PMMA) 950K A2 on 1 mm thick fused silica substrate. Then to evaporate the anisole, the sample was baked at 180 °C for 90 seconds. The array of L-shaped unit cells was patterned on the PMMA using electron beam lithography (EBL). A 30 nm copper (Cu) layer was used to eliminate the charging effect and the Cu layer was removed after exposure with nitric acid ($HNO_3$) solution and rinsing with water. After this, the sample was developed for 60 seconds in metylisobu-tylketon:IPA (MIBK:IPA) solution (1:3), and isopropanol (IPA) was used as a stopper. 1 nm Ti was deposited on the developed sample to improve adhesion between fused silica and Au and then 30 nm Au was deposited using an electron

beam metal evaporator. At last, to get rid of excess Au, lift-off was performed using S1165 remover and the sample was then cleaned using DI water.

**Numerical Simulations.** Numerical calculations were implemented by using the finite-difference time-domain (FDTD) method to verify the results. The simulations were performed by using a commercial Ansys Lumerical FDTD solutions software package. The described geometry (see Fig. 3) of L-shaped gold nanorods were placed on top of a glass substrate. The dimensions of each nanorod are $130 \times 40 \times 30$ nm$^3$ (length × width × height). The experimental complex permittivity for gold was acquired from[42] and a refractive index value of 1.45 was used for the substrate. To simulate an infinite array of these meta-molecules, the periodic boundary condition was selected (with a periodicity of 320 nm) for the $x$ and $y$ directions, while the perfectly matched layer (PML) boundary condition was used for the $z$ direction. Two different plane wave sources ($x$-polarized and $y$-polarized) were used to represent the signal and control beams, respectively. They were launched toward the structure along the positive z-axis. The signal beam has a constant amplitude of unity and a phase of 0 degrees, while the amplitude of the control beam was varied to study the effect of the control to signal field ratio (referred as $C_{amp}$). The phase of the control beam was set to $\pm\pi/2$ to induce the required phase difference. After running the simulations, the transmittance of the structure was acquired by dividing the transmitted power with the source power. The powers of both sources were obtained directly from the software, and the $x$- and $y$-polarized transmitted powers by integrating the acquired Poynting vectors at the monitor plane far below the structure. To find the $x$- and $y$-polarized components of the Poynting vectors, only the correspondingly polarized components of electric and magnetic fields were considered. To calculate the transmittance of the signal beam, the $x$-polarized transmitted power was divided with the corresponding source power. For large control beam amplitudes, transmittance values far greater than unity are observed at resonance wavelength when the phase difference is $-\pi/2$ due to the phase-controlled modulation.

Leakage of the control beam was also studied using FDTD by selecting an amplitude value of 1 for the control beam and 0 for the signal beam. The leakage was acquired as the ratio of the $x$-polarized transmitted power to the control beam source power. Only a small portion of around 3% of the source power was discovered to be leaking to the signal channel, as can be seen from Fig. 2.

**Optical characterization.** The transmission spectra of the L-shaped metasurface were recorded by exciting the sample with a broadband light source (Energetiq EQ-99XFC LDLS, spectrum from 190 nm to 2100 nm) using a microscope from WiTec (alpha300 R-Confocal Raman Imaging). The incident light is focused on the sample surface by using a 50x objective (Zeiss NA=0.75) at normal incidence. A 50x objective (Zeiss NA=0.75) is placed at the back focal plane to collect transmitted light at normal direction. Then, the collected light is coupled to an optical fiber connected to the Ocean Optics Flame spectrometer for the visible region and to the Ocean Optics NIRQuest detector for the detection of the near-infrared (800 nm - 1600 nm) range. In the incident optical configuration, a neutral density filter with an optical density of 2 (AR-coated from Thorlabs) is placed in the beam path to reduce the beam intensity so the quarter-wave plate (Thorlabs, WR 650 nm - 1400 nm) and the polarizers (Thorlabs 500 nm - 1500 nm) do not over saturate.

The measurements were performed as follows. The amplitude ratio of control to signal beams (referred as $C_{amp}$) is defined as $C_{amp} = E_y/E_x = \sqrt{I_y/I_x}$. The ratio was created by passing the broadband excitation light through a polarizer with an angle $\theta$ with respect to the $x$-axis in sample geometry. The angle was selected using

$tan\theta = C_{amp}$. For example, when an excitation beam is passed through a polarizer set at an angle ($\theta = tan^{-1}(E_y/E_x)$) $\theta = 76°$ will yield to $C_{amp} = 4$. The phase difference of $\pm \pi/2$ was induced by passing the signal and control beams through a quarter-wave plate with the fast-axis first parallel to $y$-polarization and then to $x$-polarization for the two cases. After the sample, a second linear polarizer oriented along the $x$-axis is placed to remove the control beam before the detector. The utilized polarizers are from Thorlabs and have a working range of 550–1500 nm. Thus, a longpass filter with a cut-off wavelength of 550 nm from Thorlabs was inserted before the first polarizer to remove the lower wavelengths. Before each measurement, it was verified that the desired $C_{amp}$ is indeed produced by the setup. This was done by measuring the $x$- and $y$-polarized intensities of the reference beam separately by rotating the second linear polarizer when the beam was focused on the bare transparent glass substrate without any nanoantennas. The $C_{amp}$ values were then calculated from the measured intensities, and the polarizer angle $\theta$ was optimized accordingly. The measured $C_{amp}$ values show minor fluctuations across the spectra. The measurements for different $C_{amp}$ values (i.e. $C_{amp} = 1/2, 1, 2, 3, 4, 6$) were performed in such a way that the reported $C_{amp}$ was optimized at the resonance wavelength of 800 nm.

## Data availability

Data are available from the corresponding author upon reasonable request.

## Code availability

The code used for analytical calculations in this manuscript is available from the corresponding author upon reasonable request.

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

## Acknowledgements

We acknowledge the financial support of the H2020 European Research Council (Starting Grant project aQUARiUM; Agreement No. 802986), Academy of Finland Flagship Programme, (PREIN), (320165, 320166). R.D. acknowledges the financial support of H2020 Research and Innovation Programme (Marie Skłodowska-Curie MULTIPLY Project; Agreement No. 713694). The authors thank Subhajit Bej and Petri Karvinen for their support of the e-beam lithography process.

## Author contributions

R.D. and A.P. contributed equally to this work. A.P. and H.C. developed the idea. A.P. performed the analytical calculations. R.D. and T.P. performed numerical simulations and the measurements. D.G. fabricated the samples. A.P. and H.C. supervised the project. All authors contributed to the manuscript and approved the final version.

## Competing interests

The authors declare no competing interests.
