## [Peer Review File · Nature Communications]

All-optical switching based on plasmon-induced Enhancement of Index of RefractionEditorial Note: Parts of this Peer Review File have been redacted as indicated to maintain the confidentiality of unpublished data.

REVIEWER COMMENTS

Reviewer #1 (Remarks to the Author):

The authors demonstrate that transmission amplitude of polarized light beam through a specially designed periodic plasmonic array can be altered by presence of an orthogonally polarized control beam. The physical model provided for this highly noteworthy result is the plasmon-induced enhanced index of refraction (EIR). Both the signal and control beams possess low to moderate amplitudes and observed effect does not appear to have arisen from non-linear effects, which is the most significant and unique difference with the existing literature. It seems that the authors have, for the first time, experimentally realized linear all-optical switching. The results are definitely worth immediate publication in Nature Communications, if the authors can satisfactorily support their claims by responding to the following comments:

1) The authors claim "ultra-fast" switching, however they have neither performed temporal measurements, nor they use such a light source. I wonder how come they suggest the ultra-fast nature of the phenomenon. This claim certainly requires direct evidence, otherwise it should be dropped no matter if the theoretical model suggests it.

2) In the text describing Fig 3(a), the authors state that "...signal beam's amplitude is maintained constant in all

experiments...". However in the description provided for optical measurements in section 4.3, if all the optical components used in the experiment are included in the text, then there is no such control that would allow to set signal amplitude the same in each experiment especially because of the implemented \tan_{θ} method for setting the control and signal amplitudes. The only way to keep the signal amplitude constant in all measurements is to adjust the output power of the light source, which is not mentioned in section 4.3. This point should be clarified.

3) The switching is performed via introducing phase shift in between signal and control beams, which are both supposedly incoherent beams. How can a phase-shift be defined in-between two completely incoherent beams? It seems that the reason why this experiment worked is because actually the signal and control are mutually coherent. In other words if the authors chose to use two separate light sources and combined the two beams using a beam-splitter, the experiment probably would not have yielded the same results. This point deserves an explanation and mention in the manuscript, otherwise it may lead the audience to draw wrong conclusions.

4) Did the authors also perform experiments in 3b, 4b or 5a with signal and control beams filtered by an ultra-narrow band-pass filter centered at the plasmon resonance of 800 nm? If so, do the measurements confirm broadband results? The applied amplitudes being low or moderate does not automatically discard internal non-linear conversion effects. Maybe the increase in signal transmission at 800 nm is due to non-linear plasmonic conversion. The authors should prove that it is not.

5) Where does the energy in >100% signal transmission arrive from? It must be coming from the control. But then the authors should also provide the spectral measurements of the control beam. This, for

example, is possible by using a polarizing beam splitter at the output and measuring spectra from both ends of the splitter.

Reviewer #2 (Remarks to the Author):

The paper reports on experimental validation of a very unusual effect in nanooptics, when one beam (signal beam) is enhanced by coherent transferring energy from the donor beam (control beam) with assistance of a metasurface. The remarkable thing is that such amplification happens without any gain medium. Even more, with involvement of intrinsically lossy plasmonic particles. The effect was theoretically predicted earlier, in Ref.30, and here comes the solid proof that such amplification is feasible. The core result of the whole paper is presented by two figures, Fig.3 and Fig.5, where it was shown that the growth of the signal beam at the output regarding the input part of the system reaches up to 200%. This is a very important result and thus, the paper warrants publication in the journal. However, following comments should be taken into account in preparation of the revised version.

- 1) In the abstract the authors claim that “the scheme introduces an effective tool for improving the modulation strength of optical modulators and switches through the amplification of input signals”. In general, as a theoretical speculation I would agree with this statement. Nevertheless, no any modulation modalities have been proven directly in the paper, so it can be given as an outlook for potential applications, but not as a concrete result.
- 2) Related question is that it will be of necessary importance to show how the intensity of the output signal depends on the phase shift between the beams. Certainly, it is not a straightforward option in the optical scheme used by the authors, but it can be provided at least via simulations.
- 3) The polarizability of NPs is taken through the quasistatic approximation, expression (1), which washes out any phase effects with plasmonic particles. The question is: how valid is such approach in the situation, where amplification happens due to specific phase difference between two beams (two polarizations)? Isn't it controversial?
- 4) As Fig.3 shows, the change in the phase difference between the beams from $-\pi/2$ to $\pi/2$ removes the amplified peak on the resonance frequency, but instead erects two peaks with amplification of the signal beam at lower and higher frequencies. The amplification is less than in the former case, nonetheless, from the article it is not clear, why does such shift happen.
- 5) Somewhere on page 6, a localized plasmonic resonance is called lossless. I think that this point should be elaborated as typically all plasmonic resonances are lossy in one or another degree.
- 6) Claiming “Our experimental and simulated results clearly show that this technique creates broad transparency in the complete plasmon band...” should be supported by quantitative data, because it is hard to estimate how broad is the resonance just visually from the figures. No any relevant spectral data are presented in the article.

7) Last, but not least. Complex permittivity of gold was taken from classical paper by Johnson and Christy, Ref.41. However, it is well known that parameters for thin gold films, 30 nm and thinner deviate from the bulk gold parameters. Taking into account that gold NPs thickness is about 30 nm, it is important to apply more realistic parameters of gold.

Dear Reviewers,

We are writing this rebuttal letter for our manuscript entitled “*All-optical switching based on plasmon induced Enhancement of Index of Refraction*” by R. Dhama, A. Panahpour, T. Pihlava, D. Ghindani and H. Caglayan for the publication in Nature Communications.

First, we thank you for the time and efforts devoted reviewing the manuscript. This certainly helped us to improve the value of the work and further clarify some important points in the manuscript. We also appreciate that the high level of uniqueness and novelty of this work has been recognized by the reviewers.

In the following, we report the detailed response to the comments of the two Reviewers. We have updated the manuscript with in-text modifications formatted in blue. We have also included a supplementary file to present the additional data.

On behalf of all authors,

Humeyra Caglayan

Reviewer #1 (Remarks to the Author):

The authors demonstrate that transmission amplitude of polarized light beam through a specially designed periodic plasmonic array can be altered by presence of an orthogonally polarized control beam. The physical model provided for this highly noteworthy result is the plasmon-induced enhanced index of refraction (EIR). Both the signal and control beams possess low to moderate amplitudes and observed effect does not appear to have arisen from non-linear effects, which is the most significant and unique difference with the existing literature. It seems that the authors have, for the first time, experimentally realized linear all-optical switching. The results are definitely worth immediate publication in Nature Communications, if the authors can satisfactorily support their claims by responding to the following comments:

1. The authors claim "ultra-fast" switching, however they have neither performed temporal measurements, nor they use such a light source. I wonder how come they suggest the ultra-fast nature of the phenomenon. This claim certainly requires direct evidence, otherwise it should be dropped no matter if the theoretical model suggests it.

The electronic dynamics on the surface of metal nanostructures arises due to the excitations known as surface plasmons (SPs). Due to their broad spectral bandwidth, SPs undergo ultrafast dynamics, with times as short as hundreds of attoseconds [R1, R2]. Furthermore, the relaxation rate of the surface plasmons has been reported in the literature of these articles in the 10–100 fs range across the plasmonic spectrum. This work also does not involve any slow physical processes such as thermal or slow phase change mechanisms, thus ultrafast switching is envisioned.

In particular, this manuscript highlights the significance of the plasmon analogue of EIR effect and reports all-optical switching based on the linear phenomenon at low power. Using an intense femtosecond pulsed laser to pump the L-shaped plasmonic metasurface for the investigation of the ultrafast response of the phenomenon could bring nonlinear effects into action and using ultra low energy fs pulses to avoid this could result in problems in signal detection.

However, we agree with the Reviewer on the use of word '*ultrafast*' without any temporal experiment and replaced this word with '*linear*' in the abstract. The complete sentence in the updated manuscript is as follows in the abstract.

"A linear all-optical switching mechanism is demonstrated based on a plasmonic analogue of Enhancement of Index of Refraction (EIR) effect in quantum optics."

2. In the text describing Fig 3(a), the authors state that "...signal beam's amplitude is maintained constant in all experiments...". However in the description provided for optical measurements in section 4.3, if all the optical components used in the experiment are included in the text, then there is no such control that would allow to set signal amplitude the same in each experiment especially because of the implemented \tan_{θ} method for setting the control and signal amplitudes. The only way to keep the signal amplitude constant in all measurements is to adjust the output power of the light source, which is not mentioned in section 4.3. This point should be clarified.

We realize the reason for the confusion here and accept that we should have been clearer on this. The text in the manuscript '*signal beam's amplitude is maintained constant in all experiments,*' is actually the exact description of the analytical calculations and numerical simulations where we can set the amplitude of signal beam (A_{signal}) as 1 in all calculations. For example, $C_{\text{amp}} = 4$ in the simulations (as shown in Fig. 3(c)) consists of a separate source for the signal beam which is kept constant ($A_{\text{signal}} = 1$) for all FDTD simulations. Similarly, another source is added for the control beam to set the control amplitude (A_{control}) to 4.

As the Reviewer correctly mentioned, experimentally C_{amp} is set by polarizer angle $\theta = \tan^{-1}(C_{\text{amp}})$, where C_{amp} is the ratio of E_y/E_x defined in manuscript page 7 and in section 4.3 (optical characterization) in detail. **The amplitude of signal beam is kept constant for each individual C_{amp} value with the phase difference of $\delta\phi = \pi/2$ and of $\delta\phi = -\pi/2$.** Thus, the intensity of signal beam is kept same when the phase is changed between control and signal beam for each C_{amp} value.

In order to address the Reviewer's point and make the mentioned text clearer at page No 6, line 22 "*The ratio of amplitudes of control beam to signal beam is defined as C_{amp} and signal beam's amplitude is maintained constant in all experiments, except for the case of measuring the leakage where signal intensity is removed*" has been updated in the following way:

"The ratio of the amplitudes of control beam to signal beam is defined as C_{amp} which is set by changing the input polarizer angle. For each C_{amp} value, two measurements of $\delta\phi = \pi/2$ and $\delta\phi = -\pi/2$ are performed by maintaining the C_{amp} constant."

3) The switching is performed via introducing phase shift in between signal and control beams, which are both supposedly incoherent beams. How can a phase-shift be defined in-between two completely incoherent beams? It seems that the reason why this experiment worked is because actually the signal and control are mutually coherent. In other words if the authors chose to use two separate light sources and combined the two beams using a beam-splitter, the experiment probably would not have yielded the same results. This point deserves an explanation and mention in the manuscript, otherwise it may lead the audience to draw wrong conclusions.

We agree with the Reviewer regarding the first part of the question that switching depends on the phase shift between signal and control beams. Thus, signal and control beams must be mutually coherent. It does not matter if an incoherent light source is employed to enable signal and control beam. We have already mentioned the text in the manuscript at page number 7, line 9-12 *'We note that coherent laser sources are not required to achieve coherent control of surface plasmons. Modulation strictly depends on C_{amp} and phase difference between control and signal beams which must be coherent with respect to each other.'*

In response to the Reviewer's other query to use two light sources for separate signal and control beams and insert a beam splitter to combine both beams, we would again emphasize the coherence between signal and control beams. If we can maintain the coherence, the modulation will be achieved with the phase shift between signal and control beams. For example, if we choose to use two separate incoherent light sources and combine these beams using a beam-splitter, this does not enable the coherence between both beams.

Thus, to address the Reviewer's point, we have updated and included following new text:

"We note that coherent laser sources are not necessarily required to achieve coherent control of surface plasmons. Modulation strictly depends on C_{amp} and phase difference between control and signal beams which must be coherent with respect to each other, in spite of originating from an incoherent light source. It is also noteworthy that such all-optical switching phenomenon cannot be achieved when two separate incoherent sources are employed for signal and control beams due to absence of mutual coherence."

4) Did the authors also perform experiments in 3b, 4b or 5a with signal and control beams filtered by an ultra-narrow band-pass filter centered at the plasmon resonance of 800 nm? If so, do the measurements confirm broadband results? The applied amplitudes being low or moderate does not automatically discard internal non-linear conversion effects. Maybe the increase in signal transmission at 800 nm is due to non-linear plasmonic conversion. The authors should prove that it is not.

First, we would like to clarify that we have not used any ultra-narrow band pass filter centered at the plasmon resonance of 800 nm during the experiments. However, to address the Reviewer's concern, we have repeated the measurements with the broadband response (as presented in Fig. 3b of the manuscript) and with a bandpass filter (FB800-10) as shown in Figure R1. The results clearly show that ultra-narrow responses follow the similar trends like broadband ones when the phase between control and signal beam is changed. This clearly rules out the role of non-linear plasmonic conversion in the increase in signal transmission at 800 nm.

Figure R1: Comparison in broadband and ultra-narrow response at the use of bandpass filter (800 ± 10 nm) for experimental modulated output signal intensity at $C_{amp} = 4$ as the function of phase differences of $\pm \pi/2$ between control and signal beam.

Our experimental results show rather large signal amplification (200% or more) and significant energy exchange between signal and pump channels, depending on the pump to signal power ratio. Regarding small intensity signal and pumps used in our experiments ($I < 1 \text{ W/cm}^2$), this huge energy exchange cannot be attributed to the generally weak nonlinear effects. We believe that non-linear conversion effects can be significant when intense, ultra-short laser pulses with substantial optical density illuminate the gold nanoantennas. For example, authors [R3] have measured the optical nonlinearities of gold nanoparticles by z-scan method when the sample is excited by femtosecond pulsed laser (800nm, 50fs) with intensity power at the focal point ranging from 15 GW/cm^2 – 280 GW/cm^2 . As a simple numerical estimation, considering a classical model for the nonlinearity of plasmonic nanoparticles [R4], we see that by applying low intensities on the order of 1 W/cm^2 , the contribution of nonlinear effects on the induced dipole moment of nanoparticles is approximated to be less than 10^{-6} with respect to the contribution of linear excitation.

5) Where does the energy in >100% signal transmission arrive from? It must be coming from the control. But then the authors should also provide the spectral measurements of the control beam. This, for example, is possible by using a polarizing beam splitter at the output and measuring spectra from both ends of the splitter.

As pointed out by the Reviewer, the extraordinary enhancement in output signal intensity comes from the control beam. The power flow from one nanoparticle to another or from control to signal, as visualized by finite element simulation in Fig. 7 of Ref. 30, clearly demonstrates this linear canalization of energy. This occurs in the presence of mutual coherence condition between control and signal beams through the coupled L-shaped nanoantennas. We have extensively explained the physical mechanism behind the plasmon analogue of *Enhancement of index of refraction* (EIR) effect in paragraph 3 at page 7. To avoid confusion in our explanation, we have updated the text by including the blue text to the revised manuscript *'This modification of spectral profile of LSPR and induced zero or negative absorption is attributed to the mutual coupling of both nanoantennas and the canalization of energy from vertically (control beam) to horizontally (signal beam) oriented nanoantenna in the presence of specific phase difference between signal and control beams.'*

The Reviewer is also interested in seeing the spectra of control beam and signal beam simultaneously by using a polarizing beam splitter in the detection part. However, this is not feasible in our microscope-based setup as transmitted beam collected by bottom objective is directed through two fixed mirrors to the coupler where the detected light is collected through a fiber. Thus, one fraction (transmitted one) of a beam splitter could be measured but the other fraction (reflected) of the detected beam cannot be guided to the optical path used for transmitted beam at the same time. Instead, we performed a measurement by changing the output polarizer angle to separately record the control beam.

In order to address the Reviewer's concern, we have performed FDTD simulations (Fig. R2 a) and measurements (Fig. R2 b) to evaluate the transmitted control beam intensity for $C_{\text{amp}} = 4$ with phase difference of $\delta\phi = \pi/2$ and $\delta\phi = -\pi/2$.

As shown in Fig. 3b and 3c of the manuscript, signal beam transmittance minima is observed at 800 nm with phase difference of $\delta\phi = \pi/2$. At the opposite phase difference of $\delta\phi = -\pi/2$, the substantial enhancement in signal transmission appears at the cost of control transmission as shown in Fig R2. This clearly confirms energy converting from one polarization (control beam) to another (signal beam) due to the mutual coherence condition between the beams. Note, the relative enhancement in signal transmission has been reported much higher than the decrease in control transmission. It is due to the input intensity of control beam which is significantly higher than signal input. Thus, small decrease in control intensity is enough to give rise to high signal transmittance enhancement. For example, 10% relative decrease in control transmittance can enable up to 160% increase in signal transmittance for $C_{\text{amp}} = 4$.

Figure R2: (a) Numerically simulated and (b) measured output intensity curves of the control beam normalized to control input intensity for $C_{amp} = 4$ as the function of phase differences of $\pm \pi/2$ between control and signal beam.

Reviewer #2 (Remarks to the Author):

The paper reports on experimental validation of a very unusual effect in nano-optics, when one beam (signal beam) is enhanced by coherent transferring energy from the donor beam (control beam) with assistance of a metasurface. The remarkable thing is that such amplification happens without any gain medium. Even more, with involvement of intrinsically lossy plasmonic particles. The effect was theoretically predicted earlier, in Ref.30, and here comes the solid proof that such amplification is feasible. The core result of the whole paper is presented by two figures, Fig.3 and Fig.5, where it was shown that the growth of the signal beam at the output regarding the input part of the system reaches up to 200%. This is a very important result and thus, the paper warrants publication in the journal. However, following comments should be taken into account in preparation of the revised version.

1) In the abstract the authors claim that “the scheme introduces an effective tool for improving the modulation strength of optical modulators and switches through the amplification of input signals”. In general, as a theoretical speculation I would agree with this statement. Nevertheless, no any modulation modalities have been proven directly in the paper, so it can be given as an outlook for potential applications, but not as a concrete result.

We agree with the Reviewer’s point here and changed the last line of the abstract ‘In addition, the scheme introduces an effective tool for improving the modulation strength of optical modulators and switches through the amplification of input signals.’ in the following way.

“In the pursuit of potential applications of all-optical switching devices based on linear effects, the scheme may introduce an effective tool for improving the modulation strength of optical modulators and switches through the amplification of input signals at ultra-low power,”

2) Related question is that it will be of necessary importance to show how the intensity of the output signal depends on the phase shift between the beams. Certainly, it is not a straightforward option in the optical scheme used by the authors, but it can be provided at least via simulations.

The intensity of the transmitted signal depends on the control intensity and the phase difference between the signal and control beams. However, phase shifts that result in higher modulation strength are more desirable. In order to address this issue, we have performed some analytical calculations and numerical simulations for different phase shifts and even one experiment for phase differences 0 and π . Figures R3-R5 show that phase shifts $\pm\pi/2$ and $0-\pi$ can result in higher modulation strengths (but at slightly different wavelengths) compared to other phase shift values. We also find these figures useful for the readers and Figures R3, R4 and R6 have been added to the supplementary information.

To address this question, we inserted a brief explanation to the manuscript (in the last paragraph before conclusion) about the performance of the switch for other phase differences than $\pm\pi/2$:

“Furthermore, it should be noted that the switching mechanism is not restricted to the phase differences of $\pm\pi/2$ between signal and control beams. Other phase differences may also result in significant modulation strengths, useful for switching applications, as can be seen in Fig. S2 and Fig. S3 of SI. However, the maximum modulation strength may occur at slightly different wavelengths when phase differences are changed to other values.”

Figure R3: Analytical calculations of the signal transmission when $C_{amp}=4$ and the phase difference between signal and control is $\delta\phi = \pm\pi/2$ (a), $\delta\phi = \pi$, $\delta\phi = 0$ (b), $\delta\phi = \pm\pi/3$ (c), and $\delta\phi = \pm\pi/4$ (d).

Figure R4: FDTD calculated signal transmission when $C_{amp} = 4$ and the phase difference between signal and control is $\delta\varphi = \pm\pi/2$ (a) $\delta\varphi = \pi$, $\delta\varphi = 0$ (b), $\delta\varphi = \pm\pi/3$ (c) and $\delta\varphi = \pm\pi/4$ (d).

[Redacted]

Figure R6: Signal transmission for $C_{amp} = 4$ at 806 nm (the resonance peak wavelength for $\delta\phi = -\pi/2$) for different values of $\delta\phi$ acquired via FDTD simulations. Maximum transmission of signal occurs at $\delta\phi = -\pi/2$.

3) The polarizability of NPs is taken through the quasistatic approximation, expression (1), which washes out any phase effects with plasmonic particles. The question is: how valid is such approach in the situation, where amplification happens due to specific phase difference between two beams (two polarizations)? Isn't it controversial?

At the beginning of the manuscript, we aimed at describing the behavior of the switch based on a simple analytical model with the main focus on the switching application of the EIR effect. For this purpose, we do not require a rigorous exact model for polarizability of nanoantennas similar to that in the Mie theory for spherical particles which accounts for full phase retardations and all multipolar resonances beyond dipolar. The nanoantennas under study in this work, are much smaller than wavelength and therefore quasistatic approximation is appropriate for description of the plasmon resonances.

Furthermore, even if we had used electrostatic approximation for the NPs' polarizabilities (relation (2) in the manuscript) with permittivity of metallic NPs given by the Drude formula, the phase retardation effects would not be completely washed out. We still had static depolarization or phase retardation associated with the geometrical parameter L which affects the resonance frequency of the NPs. The phase retardation effect due to Ohmic loss of metallic NPs represented by γ in the Drude formula affecting the resonance linewidth. Furthermore, even with this simpler electrostatic approximation the plasmonic EIR effect can be observed and analyzed showing reasonable conformity with simulation as demonstrated in Ref. 30.

In this manuscript, we have used the more improved quasi-static approximation which in addition to the abovementioned phase retardation effects, accounts for radiative phase retardation due to light scattering by the NPs, represented by k^3 depolarization term in relation (3), which affects the linewidth of surface plasmon resonances and consequently the overall resonance profile of the coupled NPs [R5-R7]. However, we ignored the non-radiative k^2 depolarization term because it just produces a resonance frequency red shift without affecting the overall resonance line shape. Also, our model already has reasonable conformity with simulation and experiment without accounting for this k^2 term which has no effect on the line shape in the plasmonic EIR effect.

In addition, we used the quasi-static approximation because the plasmonic nanoantenna dimensions are much smaller than the resonance wavelength and it is not necessary to take into account phase retardation effects due to higher order multi-polar resonances. In our case the inclusion of higher order resonances just complicates the analytical calculation, without introducing noticeable changes in the plasmonic EIR effect.

In order to clarify this issue and according to the above reasons we added some explanation to the first paragraph of page 3 of the revised manuscript as follows:

“Since the dimensions of nanoantennas are much smaller than wavelength in our working spectral range, dipole approximation is sufficient and there is no need for considering higher order multipolar resonances beyond dipolar. Furthermore, electrostatic approximation can be used for evaluating the polarizability of nanoantennas and description of the EIR effect as demonstrated in Ref. 30. This approximation accounts for static depolarization through geometrical parameter L and also phase retardation due to the Ohmic loss of metallic nanoantennas, represented by γ constant in Drude formula. However, here we apply the improved quasistatic approximation for achieving better compliance with simulation and experiment. In addition to the abovementioned depolarization and phase retardation effects, this approximation also accounts for radiative phase retardation due to light scattering by the nanoantennas, represented by k^3 depolarization term in the following relations, which affects the linewidth of surface plasmon resonances and consequently the overall resonance profile of the coupled NPs. We ignored the nonradiative k^2 depolarization term because it just produces a resonance frequency red shift that does not improve the conformity with simulation and experimental results.”

4) As Fig.3 shows, the change in the phase difference between the beams from $-\pi/2$ to $\pi/2$ removes the amplified peak on the resonance frequency, but instead erects two peaks with amplification of the signal beam at lower and higher frequencies. The amplification is less than in the former case, nonetheless, from the article it is not clear, why does such shift happen.

This behavior of the transmission curves due to phase change between the signal and control beams can be justified by noting that the extinction or amplification of the (x-polarized) signal is directly proportional to the imaginary part of the polarizability of nanoantennas along x direction [R2]. The real and imaginary parts of polarizability of these nanoantennas are plotted in Fig.1a of the manuscript. In this figure we see that in the case of $\delta\phi = -\pi/2$ (lower curves), the imaginary part of polarizability is strongly negative in the middle of the curve

around the resonance frequency. So, by this phase difference between the signal and pump beams we have strong amplification of signal around resonance (800nm) and extinction at lower and higher frequencies. In contrast, the upper curves in Fig.1a show that in the case of $\delta\phi = +\pi/2$, the imaginary part of polarizability is strongly positive around the resonance and rather weakly negative at lower and higher frequencies. So, we see rather high extinction around resonance and some amplification at lower and higher frequencies.

In order to justify the behavior of the transmission curves with the phase changes we updated the manuscript by adding the following explanation at page 5:

“The extinction or amplification of input signal in different spectral regions of the transmission curves are correlated with and directly proportional to the imaginary part of polarizability of nanoantennas along x direction. In spectral ranges with negative imaginary part of polarizability, amplification of signal occurs and in contrast, in the regions with positive values of imaginary polarizability, signal extinction is observed. The strength of amplification or extinction depends on the respective amount of negativity or positivity of imaginary part of the polarizability.”

5) Somewhere on page 6, a localized plasmonic resonance is called lossless. I think that this point should be elaborated as typically all plasmonic resonances are lossy in one or another degree.

We completely agree with the fact that all plasmonic resonances are lossy in nature. However, their loss can be partially compensated, either with gain medium or with coherent control of surface plasmons (mentioned in our approach). We have replaced the word ‘lossless’ with ‘low loss’ in the manuscript.

6) Claiming “Our experimental and simulated results clearly show that this technique creates broad transparency in the complete plasmon band...” should be supported by quantitative data, because it is hard to estimate how broad is the resonance just visually from the figures. No any relevant spectral data are presented in the article.

Electromagnetic induced transparency (EIT) in quantum optics and its plasmonic analogue in metasurfaces induced by coupling of the bright resonance to a narrow dark resonance, resulting in usually a narrow transparency window in contrast to broad absorption profile [Ref 27]. Such narrow band transparency is well known and most common drawback of the EIT effect which limits its practical applications in wide-bandwidth requirements.

On the other hand, in the plasmon analogue of Enhancement of Index of Refraction (EIR) effect, since both of the coupled resonances are bright states with rather large linewidths, the induced transparency also has a large bandwidth which can be further increased by enhancing the coupling strength of the two nanoantennas. As shown in Figure R7, the EIR effect enables large bandwidth transparency by modulating the absorption profile almost in the whole plasmon band of the metasurface.

We agree with the Reviewer on this point that broad transparency should be supported by the quantitative data. To address this point, we included Figure R7, showing the broad

transparency in terms of full width at half maximum (FWHM) of output signal intensity curves as the function of C_{amp} values ($C_{amp} = 3, 4$) in comparison to the plasmon band of the metasurface centered at 800 nm, to the supplementary file. For specific case ($C_{amp} = 4$), FWHM of output signal intensity curve has been mentioned by updating the text at page 9, line 2 of the revised manuscript as:

“In the plasmon analogue of EIR effect, since both of the coupled resonances are bright states with rather large linewidths, the induced transparency has also a large bandwidth which can be further increased by enhancing the coupling strength of the two nanoantennas. Our experimental and simulated results clearly show that this technique creates broad transparency almost in the whole plasmon band of the metasurface with full width at half maximum (FWHM) of 100 nm as shown in Fig. S1 of SI.”

Figure R7: Broadband transmission enhancement of signal beam in plasmon band of the metasurface as a function of amplitude of control beam (C_{amp}) with phase difference of $\delta\phi = -\pi/2$ in comparison to the intrinsic response of the metasurface when there is no polarizer and phase delay component included in optical system.

7) Last, but not least. Complex permittivity of gold was taken from classical paper by Johnson and Christy, Ref.41. However, it is well known that parameters for thin gold films, 30 nm and thinner deviate from the bulk gold parameters. Taking into account that gold NPs thickness is about 30 nm, it is important to apply more realistic parameters of gold.

Unfortunately, we lack the capability of experimentally measuring the optical constants of the used 30 nm thick gold film. To address this concern, we have compared Johnson and Christy parameters to experimentally measured parameters of 25 nm thick gold films from literature [R11]. The figure below illustrates the minor deviations of the parameters. Additionally, we ran the simulation for $C_{amp} = 4$ with both parameters and observed a slight blue-shift of the resonance with the parameters of 25 nm films (referred as Yaku 25nm). The result implies that even though neither literature parameters are an exact representation of our gold, Johnson and Christy parameters provide a better estimate when compared to the experimentally acquired response. Thus, we have demonstrated in Figure R8 that the Johnson and Christy parameters are sufficiently realistic and the simulations with them are valid to be presented in the publication.

Figure R8: a) Comparison of real (n) and imaginary (k) parts of experimental optical constants of gold from Johnson and Christy (JC) and Yakubovsky (R11, Yaku 25 nm) b) The simulated response of the structure when $C_{amp} = 4$ with constants from JC and Yaku 25nm.

References

- [R1] Stockman, Mark I. "Ultrafast nanoplasmonics under coherent control." *New Journal of Physics* 10.2 (2008): 025031
- [R2] Stockman, Mark I., et al. "Attosecond nanoplasmonic-field microscope." *Nature Photonics* 1.9 (2007): 539-544.
- [R3] Wang, Kai, et al. "Intensity-dependent reversal of nonlinearity sign in a gold nanoparticle array." *Optics Letters* 35.10 (2010): 1560-1562.
- [R4] Panasyuk, G. Y., Schotland, J. C., & Markel, V. A. "Classical theory of optical nonlinearity in conducting nanoparticles." *Physical Review Letters*, 100(4) (2008): 047402.

- [R5] Meier, M., & Wokaun, A. **Enhanced fields on large metal particles: dynamic depolarization.** *Optics letters*, 8(11) (1983): 581-583.
- [R6] Zeman, E. J., & Schatz, G. C. (1987). **“An accurate electromagnetic theory study of surface enhancement factors for silver, gold, copper, lithium, sodium, aluminum, gallium, indium, zinc, and cadmium.”** *Journal of Physical Chemistry*, 91(3) (1987): 634-643.
- [R7] Moroz, A. **“Depolarization field of spheroidal particles”.** *JOSA B*, 26(3) (2009): 517-527.
- [R8] Kuwata, H., Tamaru, H., Esumi, K., & Miyano, K. (2003). **“Resonant light scattering from metal nanoparticles: Practical analysis beyond Rayleigh approximation.”** *Applied physics letters*, 83(22) (2003): 4625-4627.
- [R9] Papasimakis, Nikitas, & Zheludev, Nikolay I. **“Metamaterial-induced transparency: Sharp Fano resonances and slow light.”** *Optics and Photonics News* 20.10 (2009): 22-27.
- [R10] Luk'yanchuk, Boris, et al. **“The Fano resonance in plasmonic nanostructures and metamaterials.”** *Nature materials* 9.9 (2010): 707-715
- [R11] Yakubovsky, D. I., Arsenin, A. V., Stebunov, Y. V., Fedyanin, D. Y., & Volkov, V. S. **“Optical constants and structural properties of thin gold films.”** *Optics express*, 25(21) (2017): 25574-25587.

REVIEWER COMMENTS

Reviewer #1 (Remarks to the Author):

I am satisfied with the revisions. The manuscript is acceptable for publication.

Reviewer #2 (Remarks to the Author):

In general, the authors have done substantial “homework” bringing some important points and clarifications to readers through the text updates and SI. Nearly all concerns brought by both reviewers have been removed. I have only one remaining. Comparing Fig. 3b,c and Fig.R2 I see that at 800 nm changes in the signal intensity about 200% by flipping the phase from $\pi/2$ to $-\pi/2$ can be attributed to decrease in the control beam intensity by 10% roughly. Then the growth of the signal beam intensity at 900 nm with the inverse phase flip, which is about 10% should be visible in the control beam output as circa 5% change. Visually I cannot support this basic arithmetic just looking in Fig.R2a,b. It will be interesting to know the authors response to this point before giving the green light for the paper publication.

Reviewer #1 (Remarks to the Author):

I am satisfied with the revisions. The manuscript is acceptable for publication.

We once again thank the reviewer and appreciate the recommendation on our revised manuscript.

Reviewer #2 (Remarks to the Author):

In general, the authors have done substantial "homework" bringing some important points and clarifications to readers through the text updates and SI. Nearly all concerns brought by both reviewers have been removed. I have only one remaining. Comparing Fig. 3b,c and Fig.R2 I see that at 800 nm changes in the signal intensity about 200% by flipping the phase from $\pi/2$ to $-\pi/2$ can be attributed to decrease in the control beam intensity by 10% roughly. Then the growth of the signal beam intensity at 900 nm with the inverse phase flip, which is about 10% should be visible in the control beam output as circa 5% change. Visually I cannot support this basic arithmetic just looking in Fig.R2a,b. It will be interesting to know the authors response to this point before giving the green light for the paper publication.

We thank the reviewer for acknowledging our successful efforts to clarify some important points through updated text in the manuscript and SI.

We also would like to mention that we believe the reviewer means 100% amplification in signal beam intensity at 900 nm with inverse phase flip instead of 10% (line 6 of the comments). There might be a typo at "... about 10% should...".

To address the reviewer’s question, we have marked the values of output control intensity at 800 nm and 900 nm as shown in Fig. R2-1. One can clearly see that in the simulation 13% decrease of the output control intensity enables up to 13% x 16 = 208 % amplification in the output signal intensity at resonance peak wavelength for $C_{amp} = 4$. The observed 206% enhancement (see Fig. 3(c) in the manuscript) is in complete agreement. At 900 nm, the change in the output control intensity is 6.5% (see Fig. R2-1a), enabling up to 6.5% x 16 = 104% modulation with inverse phase flip as mentioned in Fig. 3(c), clarifying the reviewer’s concern completely.

Figure R2-1: (a) Numerically simulated and (b) measured output intensity curves of the control beam normalized to control input intensity for $C_{amp} = 4$ as the function of phase differences of $\pm\pi/2$ between control and signal beam.

Our experimental results also report a similar trend to simulated results. The experimental data for the output control intensity has been slightly smoothed to demonstrate a change in output control beam intensity at 900 nm as shown in Fig.R2-1 (b) in comparison to Fig S1(b). The decrease in output control intensity is around 10.5 % at 800 nm, while at 900 nm a change around 4% (see Fig. R2-1b) is observed, confirming the reviewer’s arithmetic intuition. The deflection in the experimental values can be attributed to experimental inaccuracies. We have updated text in the Supplementary Information Section 1 (Spectral Response of the Control Beam) to include this discussion in the following way:

“As shown in Fig. S1(a), 13% decrease in control beam intensity can enable up to $16 \times 13 = 208\%$ amplification in output signal intensity for $C_{amp} = 4$. This is in complete agreement with 206% enhancement of output signal intensity (see Fig. 3(c) in the manuscript).”

REVIEWERS' COMMENTS

Reviewer #2 (Remarks to the Author):

Recommended for publication in its current form